# Thermally Controlled Broadband Ge$_2$Sb$_2$Te$_5$-Based Metamaterial Absorber for Imaging Applications

Zifeng Qiu [1], Gui Jin [2],* and Bin Tang [1],*

[1] School of Microelectronics and Control Engineering, Changzhou University, Changzhou 213164, China; s21060809036@smail.cczu.edu.cn

[2] Department of Electronic Information and Electronic Engineering, Xiangnan University, Chenzhou 423000, China

* Correspondence: jingui0531@xnu.edu.cn (G.J.); btang@cczu.edu.cn (B.T.)

**Abstract:** In this paper, we theoretically and numerically demonstrate a thermally controlled broadband absorber based on the phase change material Ge$_2$Sb$_2$Te$_5$ (GST). When GST operates in the amorphous state, the proposed metamaterial acts as a broadband nearly perfect absorber. The absorption can reach more than 90% in the wavelength range from 0.9 to 1.41 μm. As an application of the GST-based metamaterial absorber, the near-field imaging effect is achieved by using the intensity difference of optical absorption. Moreover, the thermally controlled switchable imaging can be performed by changing the phase transition characteristics of GST, and the imaging quality and contrast can be adjusted by changing the geometrical parameters. This designed metamaterial may have potential applications in near-infrared temperature control imaging, optical encryption, and information hiding.

**Keywords:** metamaterial; phase change materials; Ge$_2$Sb$_2$Te$_5$; perfect absorber; near-field imaging

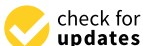



## 1. Introduction

Optical communication technology, based on electromagnetic waves [1], is particularly favored owing to its high speed and parallel characteristics. Due to the fact that electromagnetic waves can carry and transmit a large amount of information, it is beneficial for applications in optical information encryption [2–4]. In recent years, the emergence of metamaterials that can manipulate various degrees of freedom of electromagnetic waves has aroused great interest among researchers. Metamaterials are artificially engineered materials composed of periodic arrays with subwavelength scale, which exhibit various unnatural electromagnetic properties [5–8]. Since creatively proposed by Landy et al. [9], metamaterial perfect absorbers (MMPAs) have been widely studied on account of their relatively flexible design compared to traditional absorbers based on large-volume components, which have potential applications in photovoltaic, photodetectors, and imaging [10–13]. Nowadays, many types of MMPAs have been proposed, ranging from visible light to microwaves [14–16]. In particular, Y. Kivshar et al. demonstrated a plasmonic metasurface absorber supporting high-Q resonances governed by quasi-bound states in the continuum modes in the mid-IR frequency range [17]. However, most traditional MMPAs are dedicated to a single function, which cannot meet the requirements of integrated multi-function. In recent years, various tunable absorbers have been proposed by combining active materials with metamaterials, such as phase change materials [18–21], graphene [22–25], liquid crystals [26,27], and so on. For example, Li et al. utilized the toroidal dipole-bound state in the continuum to achieve perfect absorption at any desired wavelength by integrating a monolayer graphene on top of a silicon compound grating [28]. Zheng et al. proposed a tunable ultra-wideband terahertz absorber, which can achieve the switching between wideband and narrowband absorption [29]. Qi et al. proposed and demonstrated a switchable functional metamaterial device based on a hybrid graphene–VO$_2$ configuration, which can enable switching between the dual-band perfect absorption and tunable circular dichroism

(CD) response in the terahertz region [30]. Furthermore, as an alternative phase change material, $Ge_2Sb_2Te_5$ (GST) [31,32] undergoes a fast phase transition from amorphous (a-GST) to crystalline (c-GST) at around 160 °C. In particular, by precisely controlling the energy and duration of the external stimulus, GST can remain stable in an arbitrary intermediate state between a-GST and c-GST, which makes GST an ideal candidate for actively tunable optoelectronic devices. For example, Li et al. constructed a phase change metamaterial based on paired GST bars to achieve the dynamic control of the CD responses of chiral quasi-bound states in the continuum [33]. Ge et al. achieved an active control of asymmetric transmission based on topological edge states in paired photonic crystals with a-GST films [34]. Tian et al. introduced a novel MMPA based on GST, achieving an overall absorption of 92.9% across a 350–1500 nm spectrum [35]. Zhang et al. proposed a multifunctional metasurface based on GST, in which the polarization selectivity and absorption switching were achieved by controlling the phase of GST [36]. Sreekanth et al. experimentally demonstrated a GST-based absorber with multi-narrowband perfect absorption at visible frequencies [37]. In addition, there have been some exciting related works proposed for near-field imaging based on the phase change materials. For example, Gao et al. proposed a reconfigurable chiral metamaterial based on the phase change material of $VO_2$, which utilized the dual-THz band CD effect to achieve near-field imaging at 3.47 THz and 6.75 THz [38]. Xiong et al. designed and investigated a temperature tunable chirality-selective meta-absorber based on the phase change material of $VO_2$, achieving tunable near-field imaging of letters and response code with information hiding characteristics [39]. Chen et al. presented an active broadband tunable metamaterial absorber based on GST and used interferometric near-field scanning optical microscopy to investigate the near-field amplitude and phase distribution [40]. Jiang et al. introduced a GST-based metamaterial, enabling the switch between the CD intensity and display near-field images by dynamically adjusting the phase of GST [41]. However, the near-field imaging effects of current works are usually nonadjustable and difficult to be changed. Meanwhile, there are only a few studies in the literature on the realization of MMPA for switchable imaging based on GST under the excitation of linearly polarized lights.

In this paper, we theoretically investigate a thermally controlled broadband absorber based on the phase change material GST. When GST is in the amorphous state, the proposed metamaterial acts as a broadband nearly perfect absorber. The absorption is over 90% within the wavelength range of 0.9 to 1.41 μm. Then, we demonstrate the thermal control switchable near-field imaging by utilizing the absorption characteristic of the proposed absorber. Furthermore, the imaging quality and contrast can be adjustable by modifying structure parameters. The switchable near-field imaging based on GST shows potential applications in near-infrared temperature control imaging, optical encryption, and information hiding.

## 2. Structure Model and Method

Figure 1a illustrates the unit cell structure of the metamaterials, which consists of four elliptical cylinder a-GST arrays, a $SiO_2$ film, and an a-GST layer deposited on Au substrate from the top to the bottom. The thickness of $SiO_2$ film and GST layer are set to 30 nm and 290 nm, respectively. The gold substrate thickness is set to 50 nm so as to block the transmission of the incident lights. The periodicity $p$ of the unit cell in $x$- and $y$-direction is set to 1000 nm. The related geometrical parameters are listed in the caption of Figure 1. The refractive index of $SiO_2$ is set to 1.45, and the optical constants of Au are taken from the data of Palik [42]. The refractive indices of a-GST and c-GST are derived from the experimental data in reference [43]. The complex refractive index consists of a real part and an imaginary part. The real part $n$ is the refractive index of GST, while the imaginary part $k$ represents the extinction coefficient, indicating the loss of energy. It can be seen that GST has a large refractive index that is both real and imaginary, as shown in Figure 1b, which is conducive to generating strong electromagnetic absorption [44,45]. It is worth mentioning that both the real and imaginary parts of the refractive index undergo notable changes after the transition from a-GST to c-GST. The proposed structure can be fabricated using

standard deposition techniques [46] and dry etching methods, such as laser interference lithography [47].

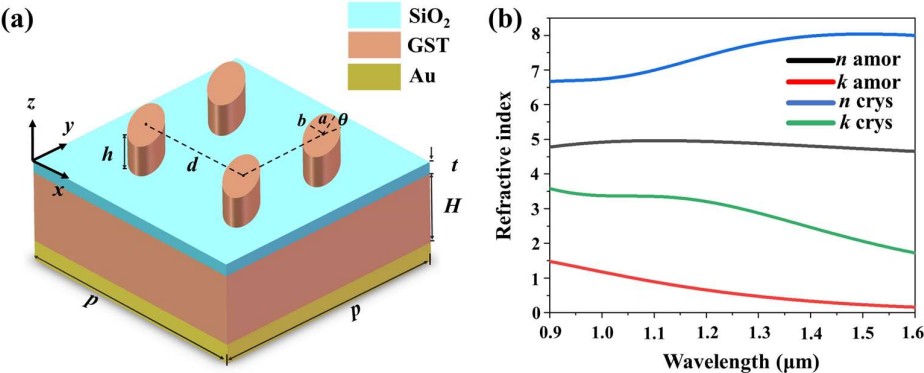

**Figure 1.** (**a**) Schematic diagram of the proposed metamaterial perfect absorber based on GST. (**b**) Real and imaginary parts of refractive index of a-GST and c-GST. The semi-major axis *a* and semi-minor axis *b* of the GST elliptical cylinder are set to 150 nm and 100 nm, respectively. *d* = 500 nm, *h* = 160 nm, *θ* = 45°, *t* = 30 nm, and *H* = 290 nm.

To study the optical properties of the metamaterials, the 3D finite-difference time-domain (FDTD) method is utilized to perform the electromagnetic responses. In simulations, the periodic boundary conditions are set in the *x* and *y* directions, and the perfectly matching layers (PMLs) are introduced in the *z* direction. The minimum mesh step is 0.25 nm and mesh type are set to auto non-uniform. The simulation time is 100,000 fs to ensure the convergence of numerical results. The electromagnetic waves are normally incident along the negative direction of the *z*-axis.

## 3. Results and Discussion

Figure 2a shows the absorption spectrum of the proposed metamaterials when GST is in the amorphous state. It is observed that the structure exhibits a high absorption efficiency exceeding 90% within the wavelength range of 0.9 to 1.41 μm. Moreover, the absorption can peak at 96% at the resonant wavelength $\lambda$ = 1.37 μm (labeled by the black point). This demonstrates that the proposed GST absorber possesses broadband and nearly perfect absorption properties when GST is in the amorphous state. The inset of Figure 2a depicts the electric field distribution of the structure in the *x*–o–*y* plane at the resonant wavelength of 1.37 μm. Then, one can see from the picture that the electric fields are mainly concentrated at the edge of GST elliptical cylinders due to the excitation of dipole resonance. As a comparison, we calculate the absorption spectrum of the metamaterials when GST is in the crystalline state, as shown by the black line. It can be seen that the absorption of the designed metamaterial in crystalline state exhibits a significant decrease. Also, one can find that the absorption difference, that is, $\Delta A = A_{a-GST} - A_{c-GST}$, reaches 0.439 at the wavelength of $\lambda$ = 1.37 μm. Figure 2b shows the absorption spectra of the metamaterial with different configurations. When the elliptical GST array is replaced by Si$_3$N$_4$ with a refractive index of 1.98, the absorption spectrum of the designed metamaterial shows a further reduction in absorption as shown by the blue line, in which the GST layer is in the crystalline state. The purple line represents the absorption of the designed metamaterial when the GST layer is in the amorphous state. Therefore, considering the significant variation in absorption spectra across different states, the designed metamaterial is intended to have the capability of being implemented on applications of near-field imaging.

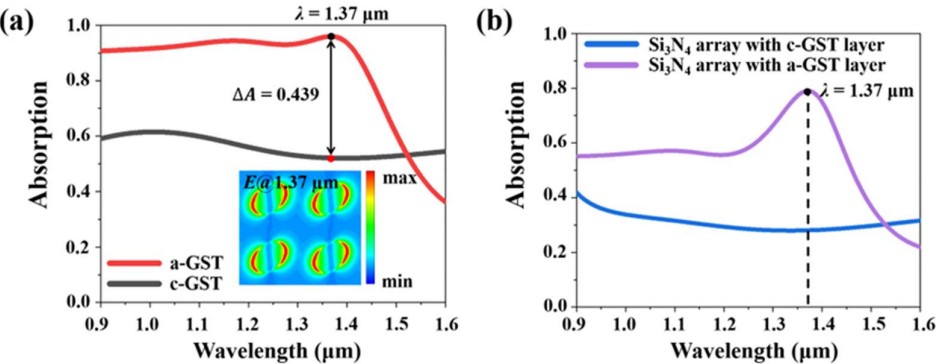

**Figure 2.** (**a**) Absorption of the metamaterial when GST is in the amorphous and crystalline states. The inset shows the electric field distribution in the *x*–*o*–*y* plane at the resonant wavelength $\lambda$ = 1.37 µm. (**b**) Absorption when the top resonator is replaced by Si$_3$N$_4$ array.

If the c-GST array and Si$_3$N$_4$ array are integrated and illuminated with incident light of a specific wavelength, the absorption difference between the two structures will cause different intensities of electric field excitation, which can be used for imaging applications. Herein, we designed a metamaterial array structure based on the GST elliptical array structure as shown in Figure 3. The proposed metamaterial array consists of 18 × 16 cells, with both the GST and Si$_3$N$_4$ elliptical arrays being identical. The GST elliptical array forms the letter "C", "Z", and "U", separately, while the Si$_3$N$_4$ array is positioned elsewhere. The initials CZU stand for ChangZhou University. The GST array and GST layer can be freely converted between a-GST and c-GST through a thermally controlled method. At the resonant wavelength of 1.37 µm, taking a plane wave incident along the *z*-direction as the source, the reflected electric field intensity is regarded as the image of the letter. The images of the letters "C", "Z", and "U" are depicted in Figure 4a–c when GST is in the crystalline state. It can be seen that the electric field of the letter is clearly distributed. When GST switches to the amorphous state, the image of the letters completely disappears (Figure 4d–f). Therefore, the proposed metamaterial array can achieve a switchable imaging function by controlling the phase transition of GST through temperature adjustments.

Figure 5 shows the image variations when the geometric parameters take different values to explore the influences of *s*, i.e., the height *h*, semi-major axis *a*, and azimuth angle $\theta$ of elliptical cylinder GST arrays. Taking the letter "C" as an example, Figure 5(a1–c1) present the images of the letter "C", with *h* being 110 nm, 160 nm, and 210 nm, respectively. It is noticeable that when GST behaves like crystals, the electric field intensity in the letter region diminishes as *h* increases. When *h* is 110 nm, the image exhibits an obvious contrast, but there are issues of color delamination and blurred edges. Additionally, the electric field distribution in the central region is uneven, which is not ideal to display in practical applications. When *h* is 160 nm, the image exhibits uniform distribution with high contrast and a clear edge, achieving the desirable imaging quality. With a further increase in *h*, i.e., *h* = 210 nm, the image display is less distinct with low contrast. Figure 5(a2–c2) reveal that when GST serves as crystals, the electric field intensity in the letter region increases with *a*. Accordingly, *a* = 150 nm is a desirable choice. Figure 5(a3–c3) show that the imaging quality gradually weakens when $\theta$ takes the values of 0°, 45°, and 90°, separately. Hence, when $\theta$ is 45°, a good balance between imaging and switchable effect can be achieved.

To quantitatively describe the imaging quality, here, we introduce the Weber contrast model [48] to describe the image contrast. It is defined as $C_w = \frac{\Delta L}{L_b}$ and $\Delta L = \mid L_t - L_b \mid$, where $L_t$ and $L_b$ represent the brightness of target and background, and $\Delta L$ is the brightness difference between the two, respectively. We set the lowest brightness of the letter to $L_t$ and the highest brightness of the background to $L_b$. Figure 6a,b indicate that as *h* and *a* increase, the image contrast rises, suggesting a gradual enhancement in the imaging quality. From

Figure 6c, it can be seen that as $\theta$ increases, the image contrast decreases. Overall, compared to its amorphous state, the imaging contrast of GST is higher when it acts as a crystal.

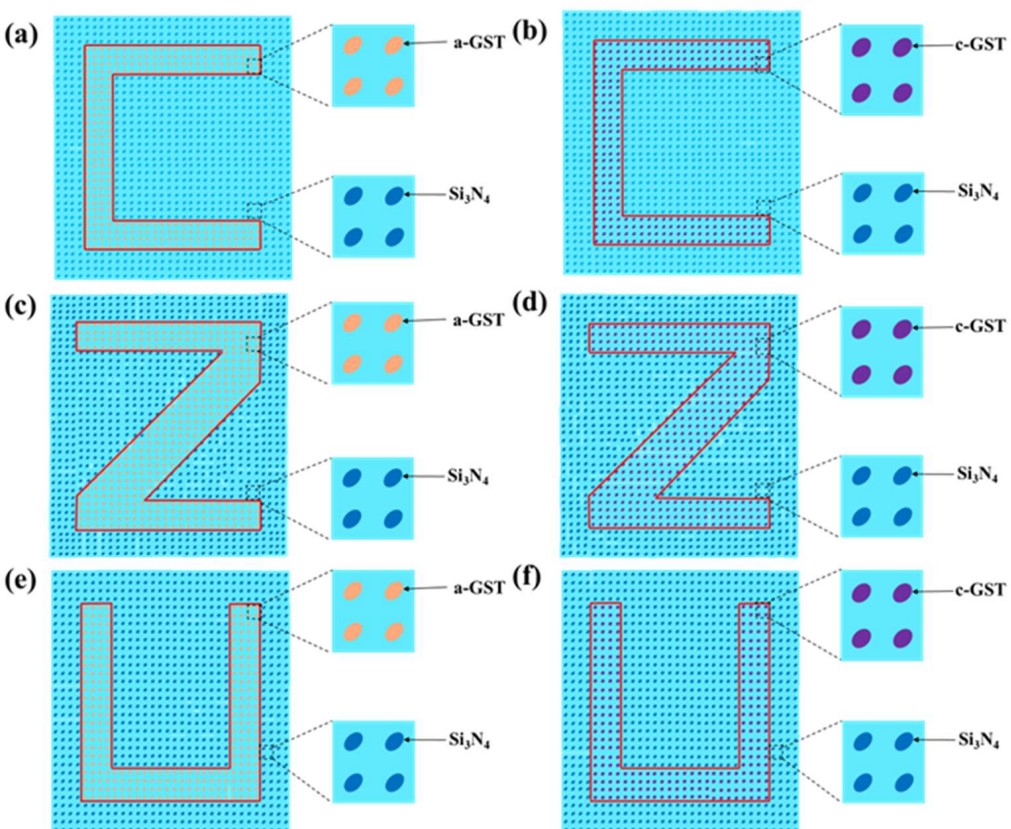

**Figure 3.** Detailed view of the a-GST (**a,c,e**) and c-GST (**b,d,f**) array arrangement of "C", "Z", and "U".

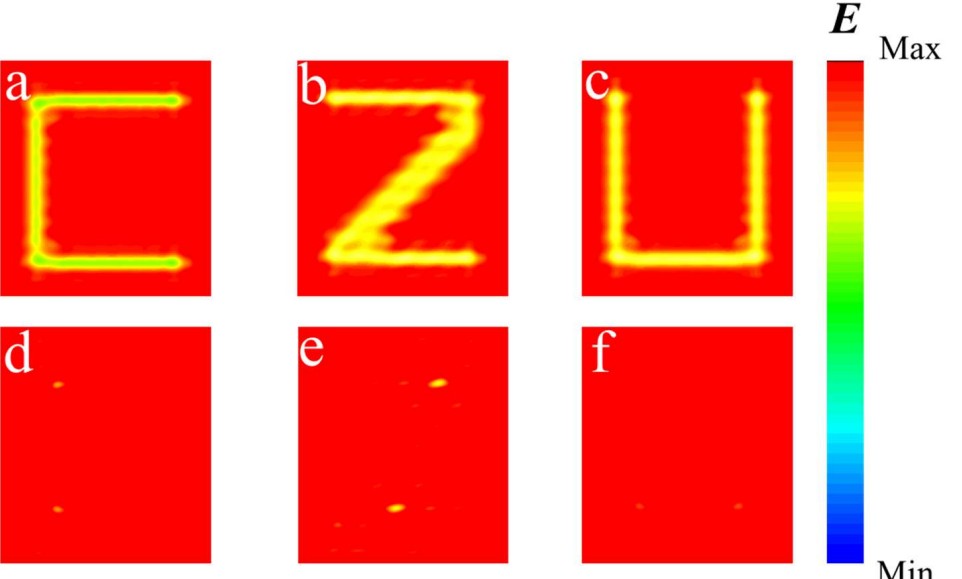

**Figure 4.** Near-field imaging of metamaterial structure with 18 × 16 cells for c-GST (**a–c**) and a-GST (**d–f**). The color bar on the far right represents an order of magnitude of electric field *E*.

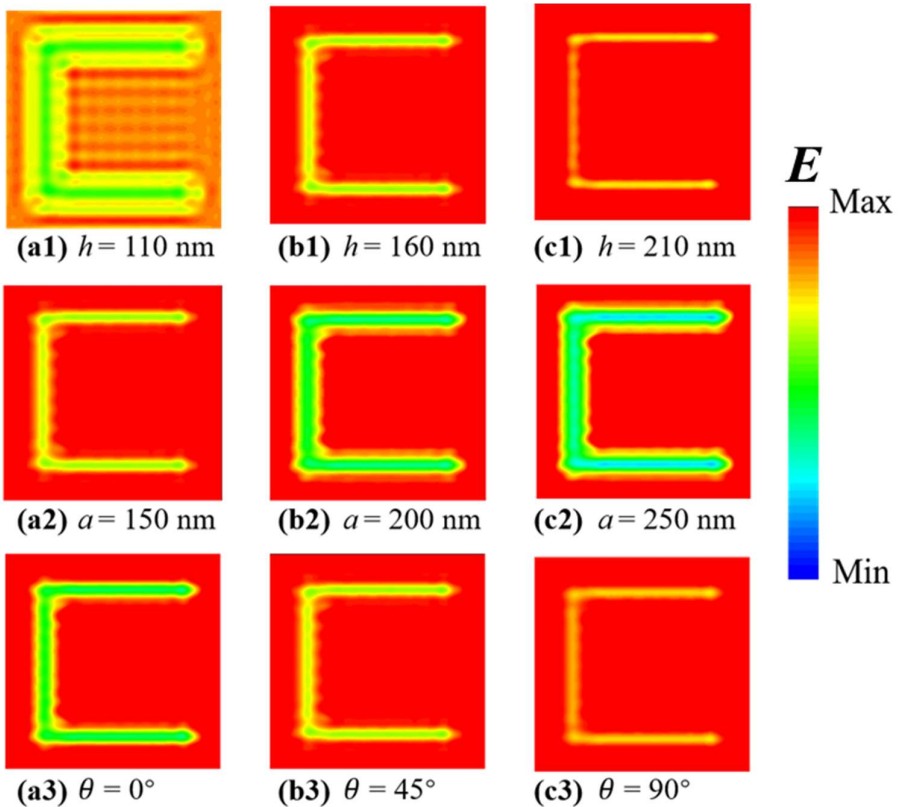

**Figure 5.** Image variations of letter "C" when the proposed metamaterial takes different geometric parameters. The color bar on the far right represents an order of magnitude of electric field *E*.

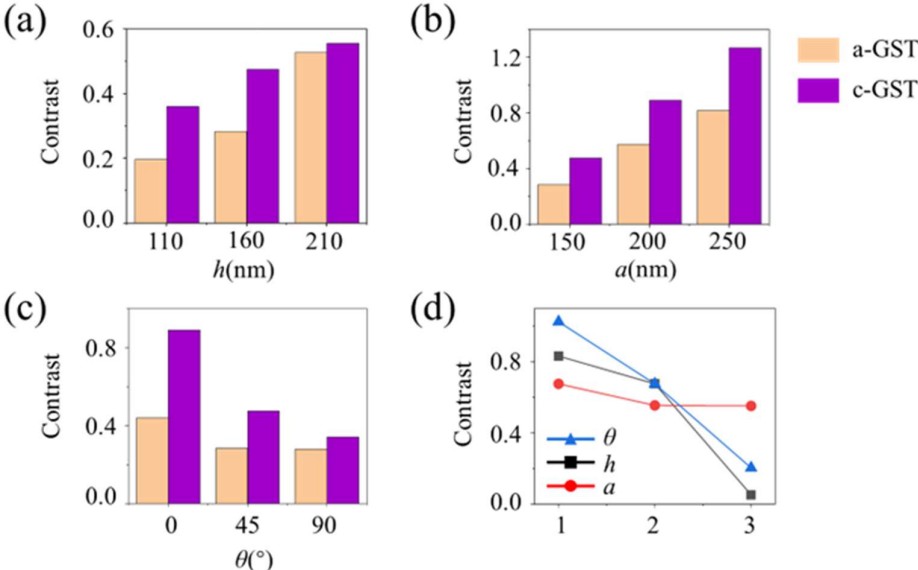

**Figure 6.** (**a–c**) Image contrast and (**d**) switchable imaging effect under geometric parameters.

In order to visualize the switchable effect, we take the contrast of the image as the "brightness", and the contrast of c-GST and a-GST are set to the target brightness $L_t$ and the background brightness $L_b$, respectively. The calculated results using the Weber contrast model are shown in Figure 6d. In this picture, "1" denotes the first set of data in Figure 6a–c, namely $h = 160$ nm, $a = 150$ nm, and $\theta = 0°$, and so on. One can observe that the imaging switchable imaging effect degenerates while the geometric parameters rise, confirming that the imaging and switchable imaging effects of Figure 5 have been numerically verified.

## 4. Conclusions

In summary, we theoretically propose and investigate a thermally controlled broadband absorber based on phase change material GST. When GST serves in the amorphous state, the proposed metamaterial acts as a broadband nearly perfect absorber with absorption exceeding 90% in the wavelength range of 0.9 to 1.41 μm. Moreover, the thermally controlled switchable near-field imaging can be achieved by utilizing the phase change characteristics and the absorption difference of different configurations. The imaging quality and contrast are adjustable by modifying the structural parameters. Our designed metamaterial may have potential applications in near-infrared temperature control imaging, optical encryption, and information hiding, etc.

**Author Contributions:** Conceptualization, Z.Q. and B.T.; methodology, Z.Q. and B.T.; software, Z.Q.; investigation, Z.Q., G.J. and B.T.; writing—original draft preparation, Z.Q.; writing—review and editing, G.J. and B.T.; supervision, B.T.; project administration, B.T.; funding acquisition, G.J. and B.T. All authors have read and agreed to the published version of the manuscript.

**Funding:** This research was funded by the applied Characteristic Disciplines of Electronic Science and Technology of Xiangnan University, and Natural Science Foundation of Jiangsu Province (BK20201446).

**Institutional Review Board Statement:** Not applicable.

**Informed Consent Statement:** Not applicable.

**Data Availability Statement:** The data are available from the corresponding author upon reasonable request.

**Conflicts of Interest:** The authors declare no conflicts of interest.

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
