# Peer review of "Thermally Controlled Broadband Ge2Sb2Te5-Based Metamaterial Absorber for Imaging Applications"

_photonics, doi:10.3390/photonics11030272_

Round 1

Reviewer 1 Report

Comments and Suggestions for Authors

In this work, the authors demonstrate a thermally controlled broadband absorber based on the phase-change metasurfaces of GST. When GST is in the amorphous state, the proposed metamaterial acts as a broadband nearly perfect absorber. Moreover, the thermally controlled switchable imaging can be achieved by changing the phase transition characteristics of GST, and the imaging quality and contrast can be adjusted by changing the geometrical parameters. In my opinion, these results are interesting and can be considered for publication after minor revision. Nevertheless, I would like to suggest that the authors should address the following main comments:

1. What was the mesh size to model the regions in the geometry? Please provide more information regarding the FDTD simulation setup.

2. Can we apply other types of phase-change materials (such as VO2) to achieve the similar absorber? What could be the possible advantages/disadvantages of using other phase-change materials? Please explain it.

3. The authors demonstrate the thermal control switchable near-field imaging by utilizing the absorption intensity differentiation at the wavelength of 1.37 µm. Can the proposed metamaterials be operated at other wavelengths?

4. Potential applications by such designed metamaterials are suggested to be discussed so as to enrich their significant importance.

5. There remains a wealth of research works into the novel functions associated with the phase-change metasurfaces of GST, such as active control of resonant asymmetric transmission [Appl. Opt. 62(22), 5969-5975 (2023)], and chiral quasi-bound states in the continuum [Opt. Lett. 48(24), 6488-6491 (2023)]. I would suggest that the authors emphasize the unique features arising from the phase-change metasurfaces of GST, which would help readers appreciate the innovative aspects of this article.

Reviewer 2 Report

Comments and Suggestions for Authors

Comments on the manuscript

The manuscript presents an active broadband absorber using GST-phase-change metamaterial for imaging applications. The absorber achieves over 90% absorption in a broad wavelength range in the near-IR regimes (about 0.9 to 1.4μm), enabling near-field imaging through optical absorption intensity differences. By altering GST phase transition characteristics and geometrical parameters, switchable imaging with adjustable quality and contrast was also studied. Potential applications include near-infrared temperature control imaging, optical encryption, and information hiding. The integration of phase change materials with metamaterials shows promise for versatile and tunable functionalities in imaging technologies.

In my assessment, the simulation/theoretical findings of this work contribute to the active metamaterial absorbers, a topic of high interest. Hence, I support the publication of this work, provided that certain concerns are addressed. Please find my suggestions and comments for the author below.

“Nowadays, many types of MMPAs have been proposed ranging from visible lights to microwaves[15-20].” Correct. However, the author did not include studies on perfect absorption using quasi-dark modes. it needs more comprehensive examples to support this claim. For example [Kivshar et al. "Bound states in the continuum in anisotropic plasmonic metasurfaces" Nano Letters, 2020; Li, et al. "Toroidal dipole BIC-driven highly robust perfect absorption with a graphene-loaded metasurface" Nano Letters , 2023].

What is the consideration of involving a thin SiO2 film? It makes the design more complicated. What if the SiO2 film is removed from your design? Please comment.

Figure 1b. The colors in this figure legend do not match the lines. For example, "k-crys". Please revise or provide clarification.

Have the authors explored the sensitivity of the imaging quality to changes in these geometric parameters and their practical implications for real-world imaging systems, considering the achieved switchable imaging effect?

Some sentences start with an “and”. For example, And one can see from the picture that the electric fields are mainly concentrated at the edge of GST elliptical cylinders due to the excitation of dipole resonance.” “And the imaging quality and contrast can be adjustable by modifying structure parameters.” Please double check and be careful.

Comments on the Quality of English Language

readable
